# The Role of Correlation in the Performance of Massive MIMO Systems

**Marwah Abdulrazzaq Naser [1], Mustafa Ismael Salman [2],*** and **Muntadher Alsabah [3]**

1   Department of Architectural Engineering, University of Baghdad, Al-Jadriya, Baghdad 10071, Iraq; marwahabdalkhafaji@gmail.com
2   Department of Computer Engineering, University of Baghdad, Al-Jadriya, Baghdad 10071, Iraq
3   Department of Electronic and Electrical Engineering, University of Sheffield, Sheffield S1 4ET, UK; mqalsabah@gmail.com
*   Correspondence: mustafa.i.s@coeng.uobaghdad.edu.iq

**Abstract:** Massive multiple-input multiple-output (m-MIMO) is considered as an essential technique to meet the high data rate requirements of future sixth generation (6G) wireless communications networks. The vast majority of m-MIMO research has assumed that the channels are uncorrelated. However, this assumption seems highly idealistic. Therefore, this study investigates the m-MIMO performance when the channels are correlated and the base station employs different antenna array topologies, namely the uniform linear array (ULA) and uniform rectangular array (URA). In addition, this study develops analyses of the mean square error (MSE) and the regularized zero-forcing (RZF) precoder under imperfect channel state information (CSI) and a realistic physical channel model. To this end, the MSE minimization and the spectral efficiency (SE) maximization are investigated. The results show that the SE is significantly degraded using the URA topology even when the RZF precoder is used. This is because the level of interference is significantly increased in the highly correlated channels even though the MSE is considerably minimized. This implies that using a URA topology with relatively high channel correlations would not be beneficial to the SE unless an interference management scheme is exploited.

**Keywords:** massive MIMO; channel estimation; spectral efficiency; spatial correlation; mean square error; ULA; URA

## 1. Introduction

The essential goal of the next generation of wireless communication systems (sixth generation (6G)) is to accommodate the significant growth in mobile data traffic [1,2]. In particular, 6G networks aim to substantially increase the spectral efficiency (SE), maintain the quality of service (QoS), and achieve a reliable communication system [3,4]. To meet these enormous demands, advanced disruptive technologies are needed [5]. One of the key enabling technologies for wireless communications systems is massive multiple-input multiple-output (m-MIMO) [6–9]. Specifically, m-MIMO systems, which utilize large-scale antenna arrays at the base station, can be used to support the tremendous growing demands in terms of data traffic [6,10,11]. The concept of m-MIMO was initially introduced by Marzetta in [10], since then, it has gained considerable attention. The advantages of m-MIMO systems are in combating the effects of the fast fading and enabling the use of linear precoding and combining techniques with a reduced signal processing complexity [12]. m-MIMO systems can also be useful for radar applications [13]. Furthermore, the array gain in m-MIMO systems can be increased significantly and the link budget can be increased proportional to the number of base station antennas. In addition, m-MIMO systems have the capability of reducing the required energy and hardware power consumption significantly [14]. From an information-theoretic point of view, the SE performance depends strongly not only on the signal-to-noise ratio (SNR) but also on the level of correlation.

Although considerable attention has been given to studying the received SNR, a little attention has been dedicated to investigating the effect of correlation. In particular, the vast majority of m-MIMO research has assumed that the channels are uncorrelated, which can be modeled using the Rayleigh fading channels [12,14–17]. These investigations rely on the assumption of the law of large numbers, which implies that when the number of antenna elements at the base station grows asymptotically large without limit, the channel coefficients become uncorrelated. Therefore, the interference can be completely eliminated. This observation is extremely strict and seems unrealistic. This is because such an interference-free scenario cannot be achieved in practice with a finite number of base station antennas. Hence, the assumption of uncorrelated channels considered conventionally by the previous research works seems highly idealistic. Furthermore, from a practical point of view, field measurements have shown that MIMO channels are correlated considering various trials and testbeds [18–23]. Therefore, investigating m-MIMO systems in a more general fading scenario, which takes into account the correlation conditions, is an interesting research topic that should be considered. This motivates us to investigate the m-MIMO system performance in terms of mean square error (MSE) and SE in a more realistic scenario where the channels are correlated.

Paper contributions and findings: The vast majority of massive MIMO research has assumed that the channels are modeled with uncorrelated Rayleigh fading channels, which seems highly idealistic. Therefore, the following contributions are offered in the present paper.

- This study addresses the challenge of investigating the performance of m-MIMO systems in a practical setting in the presence of spatial correlation and using limited coherence time.
- This study explores the impact of having arbitrary array geometries on the performance of m-MIMO systems by using different array topologies, namely uniform linear array (ULA) and uniform rectangular array (URA).
- This study seeks to answer the following question: Which array configuration maximizes the SE of m-MIMO systems. For this reason, this study considers a Laplacian physical channel model to design the channel covariance matrices with distinct array geometries. In this physical model, the correlation between the antenna elements is considered based on realistic base station settings in terms of angle of departure, angular spread, and antenna distance.
- This study considers the regularized zero-forcing (RZF) precoder technique in the downlink, which is designed based on the uplink channel state information (CSI) estimation. This is because the RZF precoder has the ability to suppress the interference when the SNR values are increased.
- This study develops an analytical closed-form expression for the MSE, which is valid for any channel model.
- Unlike state-of-the-art research works where the system performance has been investigated by using the MSE only, this study focuses on investigating the m-MIMO system performance considering both the MSE and SE with different antenna configurations.

We found that when a URA topology is used, the level of correlation is increased and the MSE is minimized. This is because the power is concentrated in a few directions under a highly correlated scenario. However, the results demonstrate that the SE is significantly degraded using the URA topology even when the RZF precoder is used. This is because the level of interference is significantly increased in the highly correlated scenario. The results show that a substantial improvement in the SE performance of the m-MIMO system can be achieved with a ULA topology in comparison to the performance obtained using a URA topology. This can be justified as follows. When a ULA topology is used, a noticeable improvement in the spatial resolution can be achieved by the array gains and the degrees of freedom in the channels, which allows the interference to be considerably reduced. However, the interference is increased when a URA topology is used because the antenna elements are closely spaced in a limited physical direction. Overall, this paper delivers

some essential insights about the m-MIMO system design, which are useful for industries and academic researchers. Finally, this paper also provides some recommendations for future work, which open up new research directions.

Paper organization: The organization of this paper is presented as follows. In Section 2, the system model is introduced. In Section 3, the effective received signal and the SE based on RZF precoding are presented. The imperfect channel estimation is discussed in Section 4. In Section 5, the physical channel correlation model is developed. To characterize the m-MIMO system performance, the numerical results are presented in Section 6. Finally, the conclusion of this research is provided in Section 7.

Notation: A bold uppercase symbol is assigned to a matrix while a bold lowercase symbol is assigned to a vector. The operation $\|\mathbf{A}\|_\mathrm{F}$ corresponds to a Frobenius norm and $\mathrm{tr}(\mathbf{A})$ denotes the trace of a matrix $\mathbf{A}$. The operations $\mathbf{A}^\mathrm{T}$, $\mathbf{A}^\mathrm{H}$, and $(\mathbf{A})^{-1}$ correspond to the transpose, Hermitian, and the inverse of matrix $\mathbf{A}$, respectively. The Kronecker product of $\mathbf{A}$ and $\mathbf{B}$ is denoted by $(\mathbf{A} \otimes \mathbf{B})$. Finally, $\mathcal{CN}(\mu, \mathbf{G})$ denotes the Gaussian distribution with mean $\mu$ and covariance $\mathbf{G}$.

## 2. System Model

In this study, we consider a scenario of a single-cell block-fading model over which the channel coefficients are timely invariant. In particular, in the typical cellular networks, the time and frequency resource blocks are allocated to different user equipments (UEs) simultaneously [24–28]. In a block-fading model, the channels are considered to be frequency flat, which remain unchanged during a constant period of time that corresponds to a channel coherence time of $T$ [29]. Two common transmission modes are used in the current generation of wireless communications systems, namely time-division duplex (TDD) and frequency-division duplex (FDD) [30–33]. The canonical m-MIMO systems are typically considered to operate in TDD mode, where the uplink and downlink transmissions share the same frequency band. TDD operation relies on the assumption that the channels in the uplink and downlink are reciprocal. In this case, the uplink channel can be used for designing the downlink precoder. Specifically, the channel reciprocity allows the uplink channel estimation to be used for downlink precoding without the requirement for downlink CSI estimation. Hence, the uplink channel estimation depends only on the number of UEs, but it is independent of the number of base station antennas $M$, where $(M \gg U)$, hence making the m-MIMO systems overhead free and fully scalable with respect to the number of base station antennas. For this reason, this study concentrates on investigating the performance of m-MIMO systems operating with TDD transmission mode only.

This study considers a base station that is equipped with $M$ antenna elements, which serves $U$ UEs simultaneously over the same time and frequency resources. Each UE is equipped with a single antenna. The UEs are perfectly synchronized. This study follows the typical assumption of an m-MIMO system, in which the number of base station antennas $M$ is much larger than the number of UEs $u$ (i.e., $M \gg U$). In each coherence block, the energy is freely divided between the useful data transmission and the pilot sequences depending on the availability of total energy at the transmitter. During the channel estimation phase, the UEs transmit UL pilot sequences of length $T_\mathrm{tr}$ with training power denoted by $\rho_\mathrm{tr}$ per channel coherence block. The remaining duration is dedicated to data transmission, which is defined as $T_\mathrm{d} = T - T_\mathrm{tr}$. During the data transmission phase, the received signal at the $u$-th UE can be expressed as

$$y_u = \mathbf{h}_u^\mathrm{H} \mathbf{d} + s_u, \tag{1}$$

where $\mathbf{h}_u \in \mathbb{C}^M$ is the complex true channel vector intended between the base station and the $u$-th UE, while $s_u \sim \mathcal{CN}(0, \sigma_\mathrm{d}^2)$ denotes the additive white Gaussian noise, which is modeled as a zero mean independent additive noise. The downlink transmit vector $\mathbf{d} \in \mathbb{C}^M$ in Equation (1) is given as

$$\mathbf{d} = \sqrt{\lambda} \mathbf{V} \mathbf{x}, \tag{2}$$

where $\mathbf{V} = [\mathbf{v}_1, \ldots, \mathbf{v}_U] \in \mathbb{C}^{M \times U}$ is the precoding matrix that is utilized at the base station to beam-form the useful data in the next transmission phase. Parameter $\lambda$ denotes the normalization constant that is used in order to ensure that the average transmit power of the base station is constant per UE during the data transmission, i.e, $\mathbb{E}[\mathbf{d}^H \mathbf{d}] = U$. As such, $\lambda$ is defined as

$$\lambda = \frac{1}{\mathbb{E}\left[\frac{1}{U}\left(\text{trace}(\mathbf{V}\mathbf{V}^H)\right)\right]}. \tag{3}$$

The following section discusses the received signal using linear precoding at the transmitter.

## 3. Received Signal Processing and Linear Precoder

This study considers that each UE knows the average effective channel through $\sqrt{\lambda}\mathbb{E}[\mathbf{h}_u^H \mathbf{v}_u]$. To this end, the received signal is given as

$$y_u = \underbrace{\sqrt{\lambda}\mathbb{E}[\mathbf{h}_u^H \mathbf{v}_u] x_u}_{\text{term1}} + \underbrace{\sqrt{\lambda}\left(\mathbf{h}_u^H \mathbf{v}_u - \mathbb{E}[\mathbf{h}_u^H \mathbf{v}_u]\right) x_u}_{\text{term2}}$$
$$+ \underbrace{\sqrt{\lambda}\mathbf{h}_u^H \sum_{l \neq u}^{U} \mathbf{v}_l x_l}_{\text{term3}} + \underbrace{s_u}_{\text{term4}}, \tag{4}$$

where the first term term1 in Equation (4) denotes the desired signal intended towards the $u$-th UE, while the second term, term2, and the third term, term3, represent the amount of interference due to the imperfect CSI knowledge and the residual inter-UE interference created by other UEs in the entire cell, respectively. The last term, term4, in Equation (4) denotes the additive white Gaussian noise, which is introduced at the UE side. It is worth noting that the interference and channel estimation error terms are neither Gaussian nor independent of the desired signal. Thus, the effective signal-to-interference-plus-noise ratio (SINR) is obtained by considering that both terms are complex Gaussian random variables and independent of the signal of interest, as considered in [34]. Therefore, the downlink SE at the $u$-th UE can be written as in Equation (5)

$$\text{SE}_u = \left(1 - \frac{T_{\text{tr}}}{T}\right) \log_2(1 + \text{SINR}_u) \quad [\text{bit/s/Hz}], \tag{5}$$

where $\text{SINR}_u$ is the effective SINR associated with the $u$-th UE, which is given as

$$\text{SINR}_u = \frac{\lambda \mid \mathbb{E}[\mathbf{h}_u^H \mathbf{v}_u] \mid^2}{\sigma_{\text{d}}^2 + \lambda \sum_{l=1}^{U} \mathbb{E}[\mid \mathbf{h}_u^H \mathbf{v}_l \mid^2] - \mid \mathbb{E}[\mathbf{h}_u^H \mathbf{v}_u] \mid^2}. \tag{6}$$

The expectation in Equation (6) is obtained by considering all sources of randomness. To this end, a Monte Carlo averaging process is used in order to compute the average SINR over random channel realizations. The channel vectors $\mathbf{h}_u$, $u = 1, \ldots, U$, are modeled as an independent complex Gaussian random distribution with zero mean and the $u$-th UE's covariance matrix, $\mathbf{G}_u$. $\mathbf{G}_u$ at the base station side is given by $\mathbf{G}_U = \mathbb{E}[\mathbf{h}_u \mathbf{h}_u^H] \in \mathbb{C}^{M \times M}$. The Gaussian random distribution accounts for the random small-scale fading realization in each coherence block. The channel covariance matrix $\mathbf{G}_u$ describes the macroscopic propagation characteristics. Clearly, the received effective signal in Equation (6) depends on the channel statistics, channel estimates, and the linear precoding technique that is used at the base station. For practical m-MIMO considerations, imperfect channel state information should be used at the base station. Hence, this study considers an imperfect CSI at the base station for precoding design. The channel estimation imperfection reduces the spectral efficiency due to the precoder mismatch with the actual channels. Therefore, it is particularly of interest to examine the system performance in terms of MSE and SE by

considering robust precoding techniques. This study considers a fully digital precoding scheme with a frequency operation below 6 GHz bands. A well-known linear precoding technique is used, which is referred to as the RZF precoder. To this end, the RZF precoder is defined as

$$\mathbf{V} = \left(\hat{\mathbf{H}}\hat{\mathbf{H}}^{\mathrm{H}} + M\phi\mathbf{I}_M\right)^{-1}\hat{\mathbf{H}}, \tag{7}$$

where $\hat{\mathbf{H}}$ denotes the imperfect channel estimation of the downlink true channel $\mathbf{H} = [\mathbf{h}_1, \mathbf{h}_2, \ldots, \mathbf{h}_U]^{\mathrm{T}} \in \mathbb{C}^{U \times M}$, which is used for data precoding at the base station. The RZF precoder uses the pseudo-inverse of the estimated channel matrix with a regularizing noise power. Parameter $\phi$ is defined as the regularization coefficient. This parameter is considered to be the inverse of the transmit SNR. As mentioned earlier, this study uses a practical channel consideration. To this end, the elements of the user channel are not isotropically distributed with an independent Rayleigh fading but rather are considered to be relatively correlated. This implies that $\mathbf{h}_u$ has dominant spatial directivity. The following section discusses the imperfect channel state information.

## 4. Imperfect Channel Estimation Analysis

In practice, channel estimation is essential for the receiver combining process and downlink data precoding. This allows the use of a simplified detection in the UL and an interference suppression in precoding. In addition, the system performance relies on the effective received signal, which depends heavily on the channel estimation. Therefore, this section presents the channel estimation process by using the practical Bayesian channel estimator. The Bayesian channel estimator uses the minimum mean square error (MMSE) filter to estimate the UL channel. The MMSE estimator makes use of noise and channel statistics to improve the accuracy of channel estimation [35,36]. In this study, we focus on the TDD transmission mode over which the channels in the uplink and downlink transmissions are assumed to be reciprocal within the coherence block $T$. Typically, the channel statistics remain constant over a specific coherence block. Each coherence transmission block consists of a number of time and frequency resources as depicted in Figure 1.

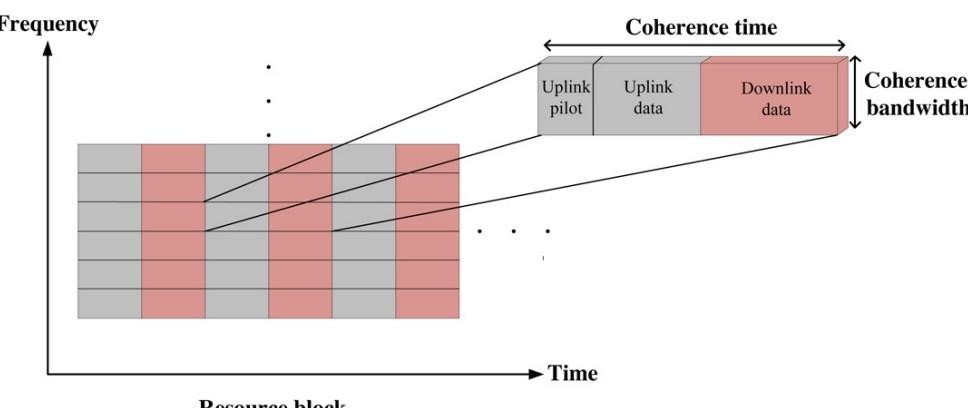

**Figure 1.** The resource block in time and frequency. The channel is assumed to be frequency flat over a specified period, which corresponds to the coherence time $T$. The pilot symbols $T_{\mathrm{tr}}$ are used in each coherence block for the uplink channel estimation in the TDD system and the rest of the resources are used for uplink and downlink data transmissions.

The base station can estimate the downlink channel by using predefined training signals that can be sent by the UEs in the uplink direction. The base station uses multiple antennas to make an accurate decision for the channel estimation by using the channel statistical distributions of the UEs. The MSE that represents the quality of the channel estimates can be minimized using the MMSE estimator. The MMSE estimator uses the received pilot signal and the known pilot book $\mathbf{P}^{\mathrm{H}}$ with mutually orthogonal sequences, which satisfy the constraint $\mathbf{P}^{\mathrm{H}}\mathbf{P} = T_{\mathrm{tr}}\mathbf{I}$. The finite length of the coherence blocks requires the sequence length to be as minimal as possible to reduce the system overhead since

longer training length comes at the cost of having a shorter period for data transmission. A rule-of-thumb is to assume the pilot length required in the uplink training is equal to the number of serving UEs in the network. The same set of orthogonal sequences is reused across the entire cell. We use a discrete Fourier transform (DFT) to generate an efficient set of mutually orthogonal sequences. An example of generating training sequences using a discrete Fourier transform (DFT) matrix is given in Equation (8):

$$
\mathbf{P} = \begin{bmatrix} 1 & 1 & 1 & \cdots & 1 \\ 1 & \omega_{T_{\mathrm{tr}}} & \omega_{T_{\mathrm{tr}}}^2 & \cdots & \omega_{T_{\mathrm{tr}}}^{(T_{\mathrm{tr}}-1)} \\ \vdots & \vdots & \ddots & \ddots & \vdots \\ 1 & \omega_{T_{\mathrm{tr}}}^{(T_{\mathrm{tr}}-1)} & \omega_{T_{\mathrm{tr}}}^{2(T_{\mathrm{tr}}-1)} & \ddots & \omega_{T_{\mathrm{tr}}}^{(T_{\mathrm{tr}}-1)(T_{\mathrm{tr}}-1)} \end{bmatrix}, \tag{8}
$$

where $\omega_{T_{\mathrm{tr}}} = e^{-j2\pi/T_{\mathrm{tr}}}$ is a $T_{\mathrm{tr}}$-th primitive root of 1. Orthogonal sequences of length $T_{\mathrm{tr}}$ are transmitted by the UEs, which allow the base station to estimate the uplink channel. A flowchart of the transmission technique is provided in Figure 2.

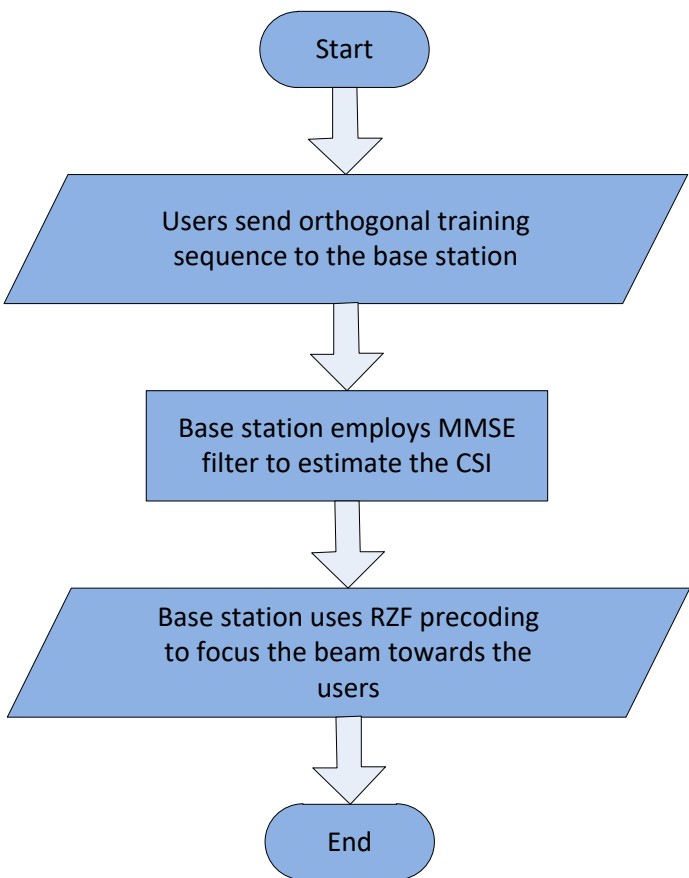

**Figure 2.** Flowchart of the transmission technique.

As discussed earlier, the uplink channel estimate is used for the downlink precoding and there is no need to estimate the downlink channel. To this end, the received training signal, $\mathbf{y}_u \in \mathbb{C}^{T_{\mathrm{tr}}}$, at the base station is given by

$$
\mathbf{y}_u^{\mathrm{tr}} = \mathbf{h}_u + \frac{1}{\sqrt{T_{\mathrm{tr}}\rho_{\mathrm{tr}}}}\mathbf{s}_u^{\mathrm{tr}}, \tag{9}
$$

where $\mathbf{s}_u^{\mathrm{tr}}$ is the receiver noise that exhibits a complex Gaussian distribution with $\mathcal{CN}(\mathbf{0}, \mathbf{I})$.

The channel vector $\mathbf{h}_u$ follows a complex Gaussian distribution where the channel statistics are known at the base station side. Recalling the received training signal

Equation (9), the channel estimate $\hat{\mathbf{h}}_u$ is obtained from the conditional probability density function (PDF) given an observation $\mathbf{y}_u$ [37]. Since the channel statistics are known at the base station, the linear minimum mean square error channel estimate [37] can be applied to estimate the channel at the base station. As such, the downlink channel estimate can be written as

$$\hat{\mathbf{h}}_k = \mathbf{G}_u \left( \mathbf{G}_u + \frac{1}{T_{\text{tr}} \rho_{\text{tr}}} \mathbf{I} \right)^{-1} \mathbf{y}_u^{\text{tr}} \tag{10}$$

$$= \mathbf{G}_u \left( \mathbf{G}_u + \frac{1}{T_{\text{tr}} \rho_{\text{tr}}} \mathbf{I} \right)^{-1} \left( \mathbf{h}_u + \frac{1}{\sqrt{T_{\text{tr}} \rho_{\text{tr}}}} \mathbf{s}_u^{\text{tr}} \right). \tag{11}$$

Using the standard MMSE estimator of Gaussian random variables with independent Gaussian noise, the covariance of the MMSE channel estimation can be written as

$$\mathbf{E}_u = \mathbf{G}_u \left( \mathbf{G}_u + \frac{1}{T_{\text{tr}} \rho_{\text{tr}}} \mathbf{I} \right)^{-1} \mathbf{G}_u. \tag{12}$$

Note that applying the MMSE channel estimate allows the $u$-th UE's channel vector to be deconstructed into the channel estimate $\hat{\mathbf{h}}_u$ and the channel estimation error $\tilde{\mathbf{h}}_u$ as shown in Equation (13).

$$\mathbf{h}_u = \hat{\mathbf{h}}_u + \tilde{\mathbf{h}}_u. \tag{13}$$

This property is achieved due to the orthogonality principle so that the vectors $\hat{\mathbf{h}}_u$ and $\tilde{\mathbf{h}}_u$ are uncorrelated. The statistical distribution of covariance matrices of the MMSE channel estimate and the covariance of the error per UE are given in Equations (14) and (15), respectively.

$$\hat{\mathbf{h}}_u \sim \mathcal{CN}(\mathbf{0}, \mathbf{E}_u) \tag{14}$$

$$\tilde{\mathbf{h}}_u \sim \mathcal{CN}(\mathbf{0}, \mathbf{M}_u) \tag{15}$$

The MSE between the actual channel and the channel estimate that can be computed through the use of Monte Carlo simulation $C_{\text{sim}}$ is given as

$$C_{\text{sim}} = \sum_{u=1}^{U} \mathbb{E} \left\{ ||\mathbf{h}_u - \hat{\mathbf{h}}_u||^2 \right\}. \tag{16}$$

Equations (12) and (16) characterize the output of a channel estimator that minimizes the MSE of each UE. The overall MSE performance relies on the transmit covariance matrix in addition to the training sequence length through $T_{\text{tr}}$ and the training power $\rho_{\text{tr}}$, which accounts for the energy used in the training stage. Algorithm 1 summarizes the algorithm used to estimate the uplink channel in a TDD massive MIMO system for a given training phase duration.

*Closed-Form Analysis for the Mean Square Error of the Channel Estimation*

In this subsection, we provide a closed-form analysis of the MSE of the channel estimate. Let the eigenvalue decomposition (EVD) of the channel covariance matrices be

$$\mathbf{G}_u = \mathbf{U}_u \mathbf{\Lambda}_u \mathbf{U}_u^{\text{H}}, \tag{17}$$

where $\mathbf{U}_u = [\mathbf{u}_{u,1}, \ldots, \mathbf{u}_{u,M}] \in \mathbb{C}^{M \times M}$ is a unitary matrix of the eigenvectors and $\mathbf{\Lambda}_u$ is the eigenvalues of $\mathbf{G}_u$ ordered as $\lambda_{u,1} \geq \lambda_{u,2} \geq \cdots \geq \lambda_{u,M}$. Using the EVD of the covariance of

the channel and channel estimate, the MSE $C_{an}$ between the actual channel and the channel estimate is obtained as

$$C_{an} = \sum_{u=1}^{U} \text{trace}\left( \mathbf{U}_u \left( \mathbf{\Lambda}_u - \rho_{tr} T_{tr} \mathbf{\Lambda}_u (\rho_{tr} T_{tr} \mathbf{\Lambda}_u \mathbf{I})^{-1} \mathbf{\Lambda}_u \right) \mathbf{U}_u^{H} \right) \tag{18}$$

---

**Algorithm 1**

---

1: **for** every user $u$ **do**
2:
3:     Calculate user distance from the BS "$D_u$"
4:
5:     **if** $D_u < 200$ **then**
6:
7:         Generate training sequence using DFT (Equation: (8))
8:
9:         Calculate the channel covariance matrix (Equations: (21) for ULA or (25) for URA)
10:
11:         Apply MMSE filter
12:
13:         Estimate the user channel at the BS (Equation: (10))
14:
15:         Compute the EVD (Equation: (17))
16:
17:         Calculate the MSE (Equation: (16) for simulation and Equation: (20) for analytical)
18:
19:         Precode the data to the users using RZF precoding (Equation: (7))
20:
21:         Calculate the spectral efficiency (Equation: (5))
22:
23:     **end if**
24:
25: **end for**

---

Noting that $\text{trace}(ABC) = \text{trace}(CAB)$ [38], hence, the overall MSE across the UEs can be further simplified into the expressions given by Equations (19) and (20), respectively.

$$C_{an} = \sum_{u=1}^{U} \text{trace}\left( \mathbf{\Lambda}_u \left( T_{tr} \rho_{tr} \mathbf{\Lambda}_u + \mathbf{I} \right)^{-1} \right) \tag{19}$$

$$C_{an} = \sum_{u=1}^{U} \sum_{m=1}^{M} \frac{\lambda_{u,m}}{1 + T_{tr} \rho_{tr} \lambda_{u,m}} \tag{20}$$

The normalized MSE (NMSE) per symbol is obtained by dividing Equation (20) into the total number of base station antennas $M$. The expression in Equation (20) can be applied for any channel covariance matrices. The expression in Equation (20) reveals that as the SNR increases, the MSE performance is improved where the estimated channel tends to be exactly the same as the actual channel. It is also explained that increasing the pilot length would enhance the MSE performance as well. Importantly, increasing the level of correlations helps in reducing the estimation error variance (the uncertainty in the channel directions) and, hence, improving the MSE performance. This is because the power in the channel will be concentrated in the strong eigendirections only and, thus, making the channel easy to estimate. Overall, the formulation in Equation (20) for the MSE is analytically convenient and, more importantly, it is straightforward to use to reproduce the numerical results. In the following section, we introduce a realistic physical channel model that is used in the performance evaluation of the TDD m-MIMO systems under consideration.

## 5. Physical Channel Model under Arbitrary Array Configurations

In practice, the channels intended for different UEs are subject to spatial correlations due to different near-field scattering environments. Measurements of propagation environments have revealed that the elements of the channel are correlated [20–22]. More

precisely, the realistic propagation environment produces more multipath components toward the serving base station from the more dominated spatial directions than from others. Therefore, to obtain a realistic propagation environment of an m-MIMO system, factors such as correlations between antennas and different scattering around the UEs should be considered. In an m-MIMO system, the antenna spacing is sufficiently small due to the large size of the array, which leads to strong correlations between the adjusted elements. This also implies that the degrees of correlation would depend on the antenna configurations and the topology of the array deployed by the base station, i.e., URA and ULA configurations. This would allow for a variation in the polarization and antenna patterns. Physically, the spatial covariance matrices are represented by the distance between the antenna elements, angle of departure, and angular spread (AS) [39]. A denser antenna array with a massive number of base station antennas may enhance the spatial resolution and improve the channel estimation accuracy. However, if the interference between the elements of the channels is very high, this might affect the received signal considerably and, thus, result in degrading the SE performance. The eigenvalue distribution of the channel covariance matrix at the base station is a metric for measuring the degrees of the correlation in the channel. For example, an identical eigenvalue distribution for all UEs is denoted by a very weak correlation or no correlation, while high channel correlations correspond to a small portion of eigenvalues that are dominated and the rest of the eigenvalues are closed to zero. In addition, strong correlations account for large eigenvalue variations.

The channel covariance matrices are subject to change on a time scale that is much slower than the coherence time. The channel gains between the base station and the UEs within the same local area can be represented by a certain correlation that stays fixed during the coherence time. This is because the channel may have the same statistics that depend on the power azimuth around the base station antenna array [40].

The eigenstructure of the channel covariance matrix $\mathbf{G}_u$ determines the spatial correlation properties of the channel vector intended for the specific $u$-th UE with a channel $\mathbf{h}_u$, which represents the geometry of the propagation paths [41]. The eigenvalues and the corresponding eigenvectors imply in which spatial directions the signal components are statistically dominated. In contrast to the conventional MIMO systems, the typical assumption in the m-MIMO literature is to have a single antenna per UE and the UEs are separable so that there is no correlations between the UEs. Extensions to multi-antenna at the UE are possible and could be considered in future work. The covariance matrix information of spatial channel characteristics is essential for channel estimation, especially when the MMSE estimator is used. In addition, the covariance matrix information can also play an essential role in resource allocation.

Methods for estimating a large dimensional covariance matrix by using a small number of observations are also possible, which can be achieved by a regularization of the sample covariance matrix [42,43]. The channel covariance matrix information can be known by the base station through the uplink channel covariance matrix [44]. Estimating the channel covariance matrix is also possible by utilizing subspace approaches, thereby avoiding the need for an instantaneous estimation of the channel [45]. An example of estimating the base station covariance matrices is provided in [46–48]. Interpolation techniques can be used [49] for obtaining the channel covariance matrices at the base station. Advanced signal processing and machine learning methods can also be exploited to achieve this purpose [50–52].

There are several approaches for modeling the spatially correlated m-MIMO channels. In the present study, we use a Laplacian physical channel model to design the channel covariance matrices of the UEs with different array geometries, i.e., ULA and URA configurations. Specifically, the Laplacian channel model is used to generate $\mathbf{G}_u$, which represents the information of the physical structure of the channel statistics. This model represents a practical scenario since it captures an arbitrary array of geometries, where the power received by each antenna varies arbitrarily, so that it experiences an unequal contribution of each antenna for a given communication link. This channel model supports both the ULA

and URA configurations. In the ULA configuration, the propagation signal happens in the azimuth direction only, while in URA configurations, the communication between the UEs and the base station happens in both azimuth and elevation directions. The physical channel model has some non-line-of-sight (NLoS) paths that correspond to the reflections from different scatterers, which produce a superposition of many paths that are jointly received. As shown in Figure 3, with the deployment of URA array topology, the parameters of the physical channel model include both azimuth and elevation angles so that the base station radiates the signals in three-dimensional (3D) space. The base station deploys the URA configuration in the y–z coordinate plane, and serves as a UE in the x–y coordinate plane. The UE channel is subject to an azimuth and elevation angle standard deviation. Each path has a corresponding angle of departure in the elevation direction $\theta_{u,E}$ and angle of departure in the azimuth direction $\theta_{u,A}$. In the horizontal ULA configuration, the signals arrive from directions within the x–y plane in the azimuth direction only along the y-axis.

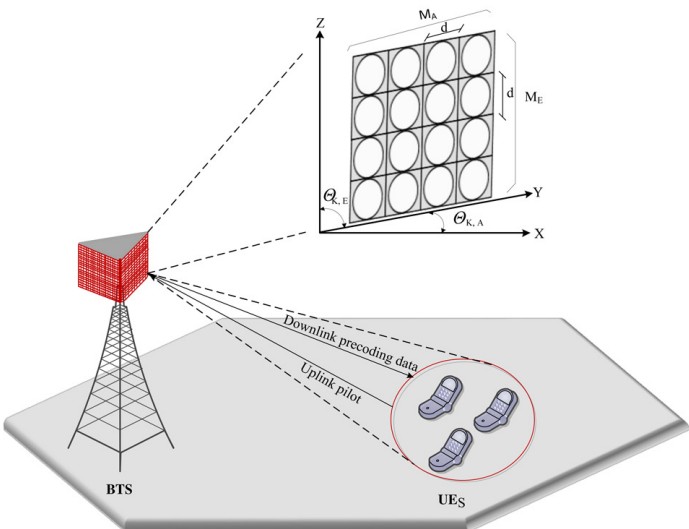

**Figure 3.** Base station deploys a URA topology, the channel model shows that the propagation link from the UEs to the base station with the azimuth and elevation angles in the coordinate plane.

In the ULA configuration, the channel covariance matrix is designed in Toeplitz form and given as

$$\left[\mathbf{G}_{u,A}\right]_{i,j} = \frac{1}{\sqrt{2}\alpha_A} \int_{-\pi+\theta_{u,A}}^{\pi+\theta_{u,A}} e^{-\frac{\sqrt{2}}{\alpha_A}(x-\theta_{u,A})} e^{\frac{-j2\pi d}{\lambda}(i-j)\sin(x)} \mathrm{d}x, \tag{21}$$

where $\alpha_A$ is the azimuth standard deviation, also known as the angular spread (AS) in the azimuth direction.

In the URA topology, the channel covariance matrix in the azimuth direction can be written as

$$\left[\mathbf{G}_{u,A}\right]_{i,j} = \frac{1}{\sqrt{2}\alpha_A} \int_{-\pi+\theta_{u,A}}^{\pi+\theta_{u,A}} e^{-\frac{\sqrt{2}}{\alpha_A}(x-\theta_{u,A})} e^{\frac{-j2\pi d}{\lambda}(i-j)\sin(x)} \mathrm{d}x, \tag{22}$$

where $\alpha_A$ is the azimuth standard deviation, also known as the angular spread (AS) in the azimuth direction.

In the URA, the channel covariance matrix in the elevation direction is given as

$$\left[\mathbf{G}_{u,E}\right]_{i,j} = \frac{1}{\sqrt{2}\alpha_E} \int_{-\pi+\theta_{u,E}}^{\pi+\theta_{u,E}} e^{-\frac{\sqrt{2}}{\alpha_E}(x-\theta_{u,E})} e^{\frac{-j2\pi d}{\lambda}(i-j)\sin(x)} \mathrm{d}x, \tag{23}$$

where $\alpha_E$ is the elevation standard deviation, also known as the angular spread (AS) in the elevation direction. The mean angle of arrival of the $u$-th UE in the vertical direction is given as

$$\theta_{u,E} = \arctan\left(\frac{\sqrt{D_u + h^2}}{h^2}\right) \tag{24}$$

The spatial covariance matrix of the $u$-th UE in the URA configuration is given in the Kronecker model as

$$\mathbf{G}_u = \mathbf{G}_{u,A} \otimes \mathbf{G}_{u,E}, \tag{25}$$

where the channel covariance matrix $\mathbf{G}_u$ is a Hermitian Toeplitz positive semi-definite matrix form.

Clearly, the correlation coefficients between the adjacent antennas at the base station side depend on angle of departure, antenna spacing ($d$), azimuth standard deviation ($\alpha_A$), and elevation standard deviation ($\alpha_E$). The channel models described above provide rank-deficient covariance matrices and the distributions of the eigenvalues are different. Note that the narrow angular spread produces a low-rank structure of the base station covariance matrix. This implies a high spatial correlation between the distinct paths that control the communication environment between the base station and UEs. In the following section, we present the numerical results in terms of MSE and SE.

## 6. Numerical Results

In this section, we provide simulation and theoretical results, which describe the system performance in terms of the normalized MSE and SE for the RZF precoder. The results are presented for the ULA and URA with realistic configurations in order to emphasize the importance of our results in a realistic channel model. A summary of the simulation parameters, which are used in the performance evaluation, is provided in Table 1. It is worth noting that 10,000 channel realizations are considered for simulating the MSE and the spectral efficiency. The angle of departure of the UEs is distributed in the range of $\theta_u \in [-\pi, +\pi)$. A dense urban scenario is assumed where the UEs are located within the range of 200 m.

**Table 1.** Simulation parameters.

| Parameters | Symbol | Value |
|---|---|---|
| Number of base station antennas | $M$ | 16, 64, 128, 256 |
| Number of UEs | $U$ | 10 |
| Base station height | $h$ | 35 m |
| Number of training slots | $T_{tr}$ | 10 symbols |
| Azimuth standard deviation | $\alpha_A$ | $5°, 10°, 15°, 20°$ |
| Elevation standard deviation | $\alpha_E$ | $2°$ |
| Antenna distance spacing | $d$ | $\lambda/2$ |
| UEs' location range from BS | $D_u$ | $<200$ m |
| Coherence time | $T$ | 100 symbols |

### 6.1. Performance Evaluation Based on the MSE

In this subsection, we compare the MSE performances under different antenna configurations. Figures 4 and 5 show plots of the MSE in the ULA and the URA, respectively, versus the SNR values in a scenario where the azimuth standard deviation is $\alpha_A(AS) = 5$. The lines depict the numerical analysis based on Equation (20), while the colored markers denote simulation based on Equation (16). Clearly, an excellent agreement between the analytical and simulated results is obtained with the realistic channel model.

Figures 4 and 5 demonstrate that increasing the number of base station antennas results in reducing the MSE of the channel estimate. The plots also illustrate that the MSE of the channel estimate is minimized using an array with a URA configuration. For

example, at $M = 128$, to achieve an MSE of 0.001, 12.5 dB transmitted power is required in the ULA while it required only 7.5 dB with the URA topology.

Figures 6 and 7 show plots of the MSE in the ULA and the URA, respectively, versus the SNR under different correlation coefficients in terms of the azimuth standard deviation $(\alpha_A)$ AS in degrees.

Figures 5 and 6 demonstrate that when the azimuth standard deviation increases, the MSE performance is increased. This is because the degrees of the correlations in the channel are relatively increased when the azimuth standard deviation is increased. In particular, a significant improvement in the MSE performance is achieved when the azimuth standard deviation is reduced, which means that the degrees of correlations in the channel are relatively increased.

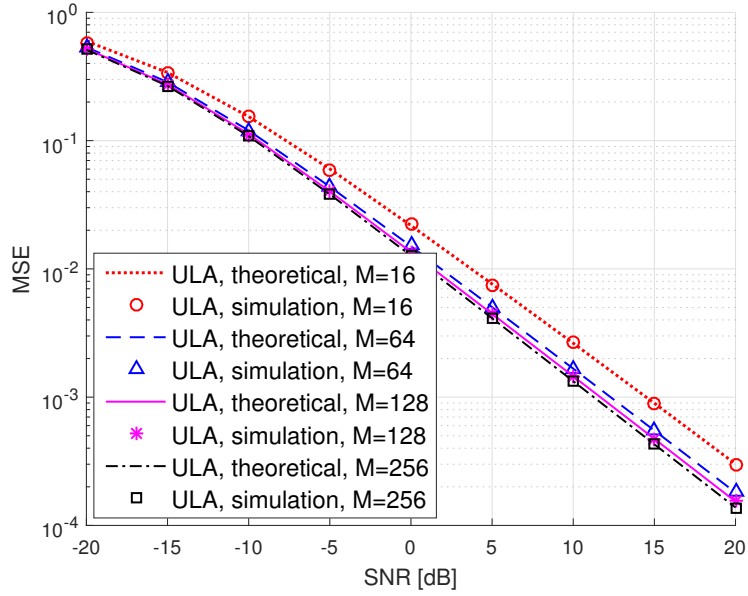

**Figure 4.** MSE vs. SNR in dB for the ULA topology, $U$ = 10 UEs, $\alpha_A(AS) = 5$, with different numbers of base station antennas.

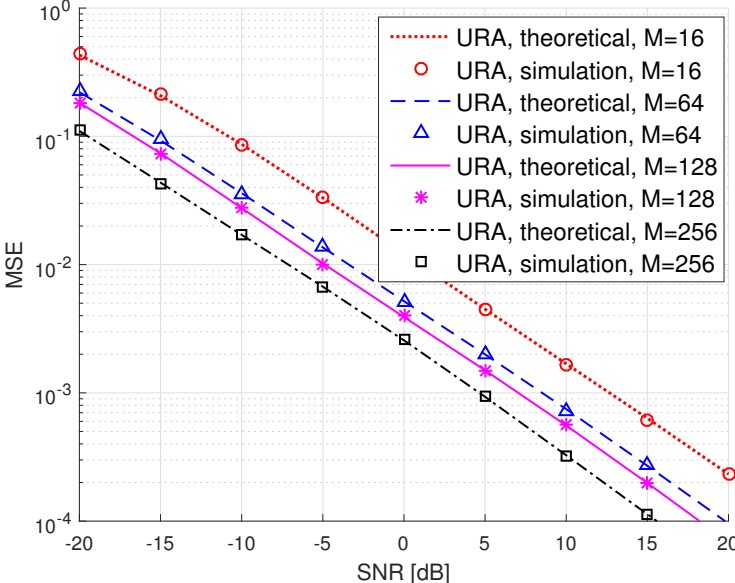

**Figure 5.** MSE vs. SNR in dB for the URA topology, $U$ = 10 UEs, $\alpha_A(AS) = 5$, with different numbers of base station antennas.

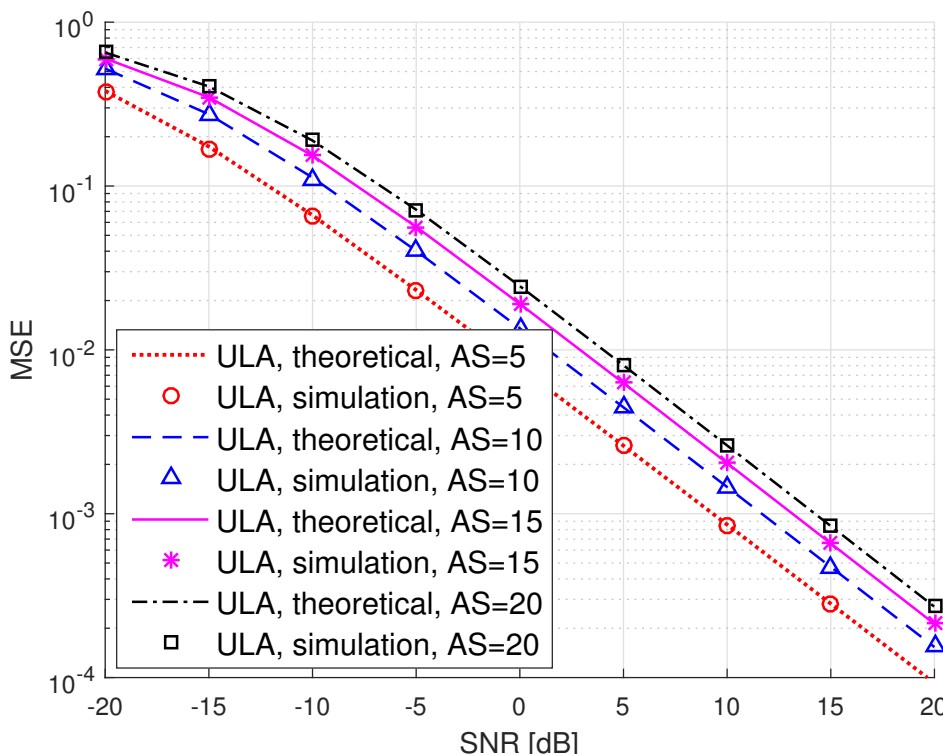

**Figure 6.** MSE vs. SNR in dB for the ULA topology, $U = 10$ UEs, $M = 128$, with different values of angular spread $\alpha_A(AS)$.

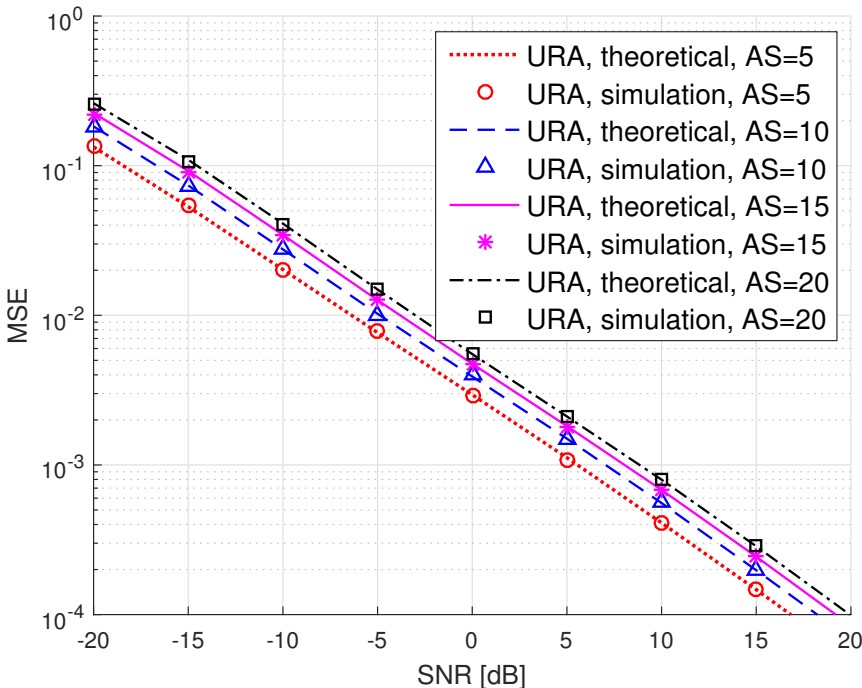

**Figure 7.** MSE vs. SNR in dB using the URA topology, $U = 10$ UEs, $M = 128$, with different values of angular spread $\alpha_A(AS)$.

Conclusions from the MSE figures can be drawn as follows: as the transmit power increases, the estimation error decreases since the error variance approaches zero in the asymptotic power regime. Another important insight is that the error variance is reduced with the stronger eigendirections, which correspond to the largest eigenvalues of the

covariance matrices of the UEs. This is explained by the fact that the transmit power is relatively high, which is concentrated in the stronger eigendirections of the channel, thus improving the performance of channel estimation significantly.

### 6.2. Performance Evaluation Based on Spectral Efficiency

So far, the results have been presented in order to evaluate the system performance based on MSE performance. This section evaluates the system performance considering pectral efficiency.

Figures 8 and 9 examine the spectral efficiency performance for the RZF precoding technique vs. the SNR in dB for the ULA and URA topologies, respectively. In particular, Figure 8 shows that at a typical SNR value, i.e., 10 dB, the spectral efficiency is increased by almost 50 bit/s/Hz when the number of service antennas is increased from $M = 16$ to $M = 256$. However, the performance gap is reduced when the number of service antennas is increased to above $M = 64$. Furthermore, the results in Figure 8 show that a 15 dB gain is achieved by using $M = 256$ in comparison with $M = 16$ using the same transmit power. Interestingly, to achieve the same spectral efficiency performance, we can increase the number of service antennas from $M = 16$ to $M = 64$. As such, a 10 dB transmit power gain is achieved by adding extra service antennas. This is a reasonable price to pay for saving the transmit power. As can be observed from Figures 8 and 9, the spectral efficiency benefits significantly from increasing the number of base station antennas. This is due to the fact that the signal intended for different UEs is spatially focused when the number of antennas at the base station increases, which results in a significant improvement in spectral efficiency. Clearly, deploying the ULA topology at the base station achieves a considerable gain in the spectral efficiency in comparison with the URA-based configuration.

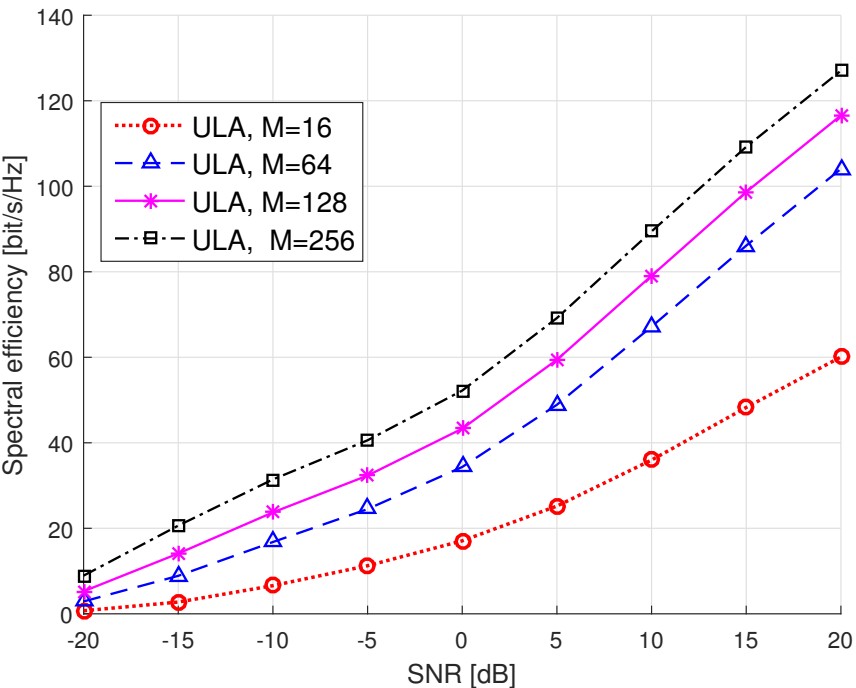

**Figure 8.** Spectral efficiency vs. SNR in dB for the ULA topology, $U = 10$ UEs, $\alpha_A(AS) = 5$, with different numbers of base station antennas.

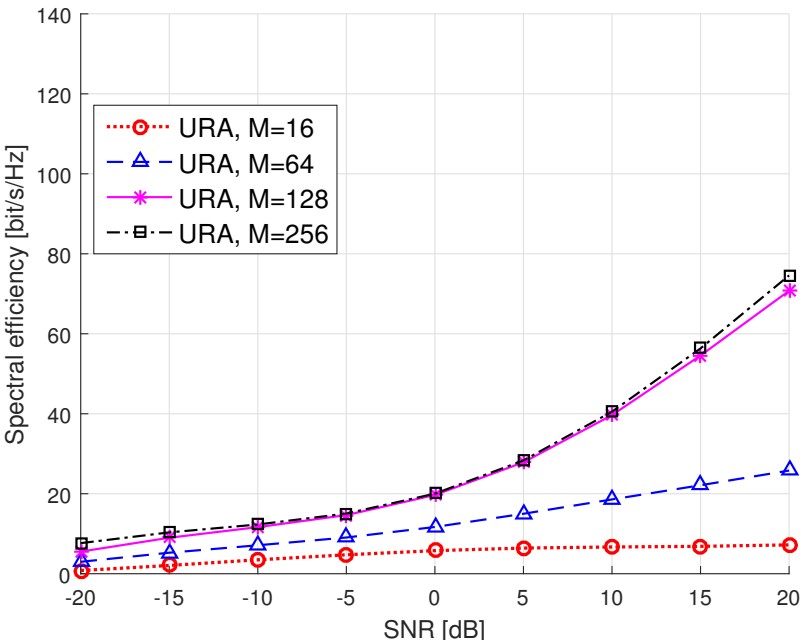

**Figure 9.** Spectral efficiency vs. SNR in dB for the URA topology, $U$ = 10 UEs, azimuth standard deviation $\alpha_A(AS) = 5$, with different numbers of base station antennas.

Figures 10 and 11 plot the spectral efficiency vs. the number of base station antennas $M$, for the ULA and URA configurations, respectively, comparing different correlation levels by using various standard deviations $\alpha_A$ (AS). An important insight that can be drawn from the plots in Figures 10 and 11 is that increasing the level of correlation by decreasing the azimuth standard deviation $\alpha_A$ (AS) results in reducing the spectral efficiency. In addition, deploying the ULA topology at the base station results in improving the spectral efficiency significantly in comparison with the URA-based configuration. This is due to the fact that increasing the interference between the covariance matrices of the UEs would lead to reducing the received signal strength and, thus, reducing the spectral efficiency performance.

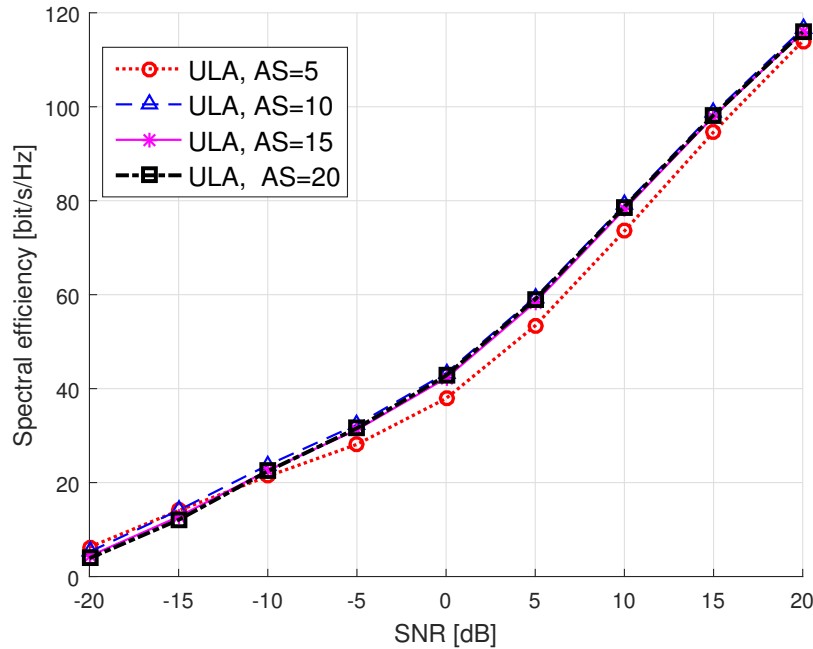

**Figure 10.** Spectral efficiency vs. SNR in dB for the ULA topology, $U$ = 10 UEs, $M = 128$, with different values of angular spread $\alpha_A(AS)$.

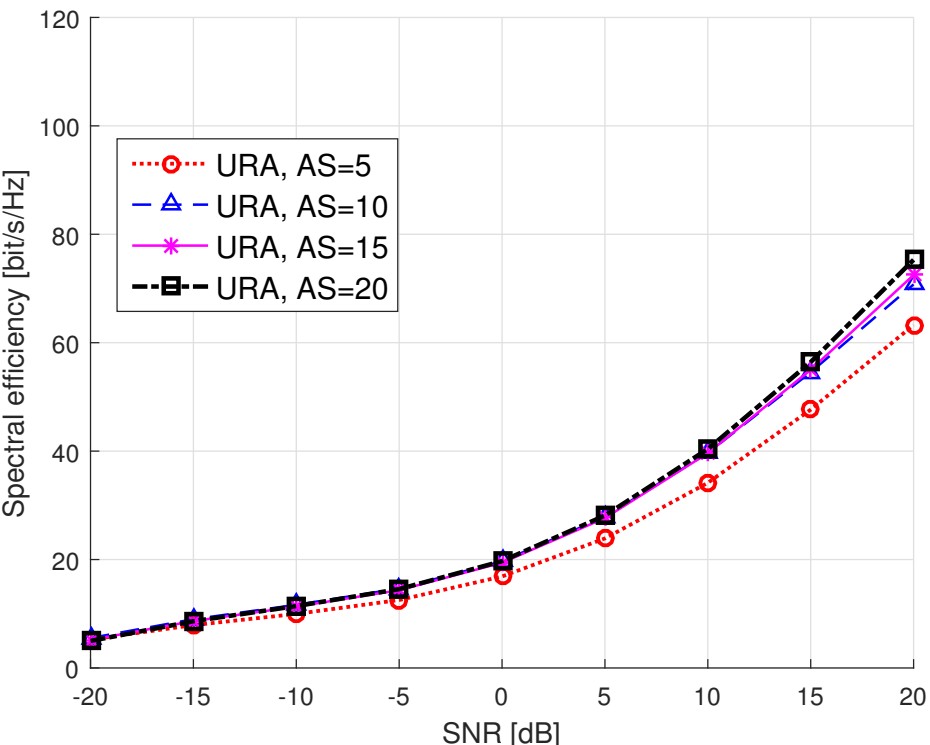

**Figure 11.** Spectral efficiency vs. SNR in dB for the URA topology, $U$ = 10 UEs, $M$ = 128, with different values of angular spread $\alpha_A(AS)$.

The analysis of the results provides a clear insight into the operation and interference characteristics of an m-MIMO system when different levels of correlations at the base station are considered. Overall, we identify a set of scenarios under which the m-MIMO system can achieve a minimum MSE and a maximum SE. The results confirm that minimizing the MSE of the channel estimate would not necessarily maximize the SE for the m-MIMO systems under consideration.

## 7. Concluding Remarks and Future Research Directions

This study evaluated the m-MIMO performance using different levels of correlations at the base station and considering different topologies with ULA and URA configurations. To this end, these different topologies and their resolution capabilities for estimating the channels were examined. The results showed that increasing the correlations minimizes the MSE of channel estimates. This MSE enhancement comes from reducing the uncertainty in the channel, which needs to be estimated. However, this enhancement is not necessary for improving the SE performance. This is because increasing the correlation would increase the interference in the channels and, thus, reduce the overall SE. As such, there is a requirement for developing a robust interference management technique in order to avoid SE degradation. In addition, this study showed that there is a trade-off between the MSE minimization and the SE maximization. This trade-off should be carefully considered when making a decision on which antenna configurations are used. Although this study considers frequency-flat channels, frequency-selective channels with orthogonal frequency-division multiplexing (OFDM) can be considered in future. Furthermore, considering m-MIMO systems with a terahertz (THz) frequency is also worth investigating in future. Finally, evaluating the m-MIMO using a hybrid automatic repeat request for spatial multiplexing transmission systems, as in [53], may also be considered in future.

**Author Contributions:** Conceptualization, M.A.; Data curation, M.A.N.; Formal analysis, M.A.N. and M.I.S.; Funding acquisition, M.A.N. and M.I.S.; Investigation, M.A.N.; Project administration, M.A.N.; Resources, M.A.; Software, M.A.; Validation, M.I.S.; Visualization, M.A.N.; Writing—original draft, M.A.N.; Writing—review and editing, M.I.S. and M.A. All authors have read and agreed to the published version of the manuscript.

**Funding:** This research received no external funding.

**Institutional Review Board Statement:** Not applicable.

**Informed Consent Statement:** Not applicable.

**Data Availability Statement:** Not applicable.

**Acknowledgments:** The authors would like to acknowledge University of Baghdad for general support.

**Conflicts of Interest:** The authors would like to declare that there is no conflict of interest.

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
