# Peer review of "The Role of Correlation in the Performance of Massive MIMO Systems"

_asi, doi:10.3390/asi4030054_

Round 1

Reviewer 1 Report

In this work, the impact of having arbitrary array geometries on the performance of the m-MIMO systems is investigated in terms of minimum MSE of channel estimation and maximum spectral efficiency. Both uniform linear array (ULA) and uniform rectangular array (URA) topologies are evaluated. Their results are interesting, and the contribution of the work is marginal for possible publication in this journal. However, it needs some improvements, as follows:

1- In Page 2, K is not clearly defined. If it is the number of UEs, it is then defined as U.

2- The author should clearly define the targeted operating frequency in this work. The expression in the last line of Page 4 is not appropriate that says "this paper does not specify a particular frequency range ,... ". What is aimed in this work seems to be more important for sub-6 GHz (as considered by the authors), since the propagation channel is different in for example mmWave which results in different levels of the spatial correlation.

3- The explanation before Eq 11 is confusing for the readers since E_u is covariance of the channel estimate, and there is another term (h-tilde) which is defined as the estimation error.

4- There are a few typos and grammatical errors. For example, in the last paragraph of Page 7: "angler spread", in the second paragraph of Page 8: "... that is stay fixed ... ", in the third paragraph of Page 8: "... so that there no correlations between ... "

5- In Numerical Results, please provide more information about the simulation. For examples, how many runs have been considered for MSE.

6- In Section 6.1, the discussion provided for Figures 5 and 6 is not consistent with what is shown in the figures. The MSE increases for higher azimuth standard deviation.

7- In the caption of Fig. 8, the azimuth standard deviation is shown with α_H while it is defined as α_A in the manuscript.

8- How about the analysis of elevation standard deviation in URA topology? It can be interesting.

Reviewer 2 Report

The paper evaluated the impact of having arbitrary base station antenna array geometries on the performance of massive MIMO systems. it is an interesting paper to read, suggested changes are:

  1. please add page numbers in
  2. what is the different between 5G and 6G? what is the role that m-MIMO play in these two wireless network?
  3. Please check if Equation 4 is correct.
  4. Some terms that used should be given the full name, i.e BS, DL, UL etc.
  5. Table 1, please state where the figures in the Table 1 comes from?
  6. it would be useful to included the experimental test data so you can compare the results with your simulation results
  7. Figure 7, there are quite big different with results presented for spectral efficiency, perhaps some further explanation on the results would be helpful.
  8. Concluding remarks and future research directions need to be more clearer, suggest to rewrote this section.

Reviewer 3 Report

Authors have highlighted the emerging and core issue, but still there are major issues to be fixed.

Reviews to Authors

  • Title must be simple, clearer and nicer.
  • Spell out each acronym the first time used in the body of the paper. Spell out acronyms in the Abstract by extending it.
  • The abstract can be rewritten to be more meaningful. The authors should add more details about their final results in the abstract. Abstract should clarify what is exactly proposed (the technical contribution) and how the proposed approach is validated.
  • What is the motivation of the proposed work?
  • Introduction needs to explain the main contributions of the work clearer.
  • The novelty of this paper is not clear. The difference between present work and previous Works should be highlighted.
  • Authors must explain in detail the introduction section.
  • Authors must develop the framework/architecture of the proposed methods
  • There is need of flowchart and pseudocode of the proposed techniques
  • Proposed methods should be compared with the state-of-the-art existing techniques
  • Research gaps, objectives of the proposed work should be clearly justified.
  • For strengthening the Introduction and related work sections authors are highly recommend to add these high quality works <‘A Compact High-Gain Coplanar Waveguide-Fed Antenna for Military RADAR Applications’, International Journal of Antennas and Propagation, Vol. 2020, Article ID 8024101, pp.1-10, 2020 >, <"HARQ with chase-combining for bandwidth-efficient communication in MIMO wireless networks,"2018 International Conference on Computing, Mathematics and Engineering Technologies (iCoMET), Sukkur, Pakistan, pp. 1-6, 2018>.

  • English must be revised throughout the manuscript.
  • Limitations and Highlights of the proposed methods must be addressed properly
  • Experimental results are not convincing, so authors must give more results to justify their proposal.
  • Table for simulation parameters must be added

Finally, paper needs major improvements

Round 2

Reviewer 2 Report

The authors have made the changes as the reviewers suggested, no further changes as required.

Author Response

Thanks for your valuable comments. 

Reviewer 3 Report

Authors have improved the paper significantly, so I recommend minor changes 

Reviews for the Authors

  1. it is suggested to write the contribution into bullets form with detail, because it shall give the quick insight instead of going to find it after reading long text
  2. Pseudocode of the proposed method is missing, so authors are recommended to provide it.

Minor changes are required
